# Reusing Discarded Ballast Waste in Ecological Cements

**DOI:** 10.3390/ma12233887

**Published:** 2019-11-25

**Authors:** Santiago Yagüe García, Cristina González Gaya

**Affiliations:** ETS Ingenieros Industriales, Universidad Nacional de Educación a Distancia (UNED), C/Juan del Rosal, 12, 28040 Madrid, Spain; cggaya@ind.uned.es

**Keywords:** cements, ballast waste, cornubianite, mechanical properties, Spain

## Abstract

Numerous waste streams can be employed in different cement production processes, and the inclusion of pozzolans will, moreover, permit the manufacture of concrete with improved hydraulic properties. Pozzolanic materials can be added to Ordinary Portland Cement (OPC) in the range of 10%–20% by mass of cement. One such example is the phyllosilicate kaolinite (K), and its calcined derivative metakaolin (MK), incorporated in international cement manufacturing standards, due to its high reactivity and utility as a pozzolan. In the present paper, discarded ballast classed as Construction and Demolition Waste (C&DW) is reused as a pozzolanic material. Various techniques are used to characterize its chemical, mineralogical, and morphological properties, alongside its mechanical properties, such as compressive and flexural strength. Discarded ballast in substitution of cement at levels of 10% and 20% produced type II or IV pozzolanic cements that yielded satisfactory test results.

## 1. Introduction

The cement industry is one of the main contributors of greenhouse gas emissions, such as CO, CO_2_, and NO. In this sector, new strategies are now prioritized in many countries that are trying to surpass the European Union (EU) targets set for 2020, which aspire to 20% lower emission levels than in 1990. Natural additions to cement have been used for the improvement of its properties since the days of antiquity. One of the best known is the addition of pozzalonic material consisting of a type of volcanic pumice, purportedly found around the town of Pozzuoli (Italy) that causes certain hydraulic characteristics in cement, hence the name of pozzalonic cements [1].

A pozzolan material is understood to be a material that generally consists of silica and alumina that, in itself, has no cementitious value when mixed with water. However, when finely ground and in the presence of water, it reacts with portlandite (Ca(OH)_2_), originating from the hydration of silicates present in the clinker, generating compounds with cementing properties [2]. The normalized artificial additions that present pozzolanic behavior are industrial by-products, such as silica fume and fly ash—the natural pozzolans—and calcined bauxite. The non-normalized materials include paper sludge, calcined sugar cane bagasse ash, and rice husk ash, among others.

Various natural materials that might function as pozzolans have been tested in cement research, and various artificial additions have more recently been undergoing trials, especially industrial waste materials including slags, silica fume, fly ash, etc., which are usually dumped in large volumes in landfill sites [3]. The utility of their addition to cement is two-fold: the elimination of waste and the enhancement of the cement. These additions are understood as technological solutions, designed to improve both the performance of high-strength cements and cement behavior against aggressive environmental agents [4]. Hence, the incorporation of these additions to cement can be seen as an attempt to reduce manufacturing costs, while simultaneously searching for materials that are more respectful towards the environment.

In the current panorama of waste recycling, construction and demolition waste (C&DW) assumes fundamental importance, because it constitutes one of the main waste streams within the EU. The recycling/reutilization rates of C&DW in the EU vary significantly between countries, fluctuating between 5% in Portugal, and 90% in the Netherlands and Estonia. On average, the recycling rate in the 27 countries of the EC is 55% [5].

According to data from the Plan Nacional de Residuos de Construcción y Demolición 2008–2015 (National C&DW Plan in Spain), 40 million tons of C&DW are produced annually, which is equivalent to over 2 kg per person, per day—a higher rate than domestic rubbish. From data provided by the Environment Ministry, 2.5 million tons were recycled in 2018. It implies a C&DW recycling rate of 5.1%—a figure well below the European average—implying that the main means of disposing of these wastes continues to be in regulated or unregulated landfill sites in Spain, as opposed to their recycling or reuse.

The publication, in February 2008, of Royal Decree (RD) 105/2008, in regulation of the production and management of C&DW (BOE of 13 February), implies growing environmental concerns and interest in this matter, that national government and the autonomous regions have been expressing for numerous years. Prior to this point, there was the 2002–2006 Plan Nacional de Residuos Urbanos (PNRU) [National Urban Waste Plan] and, more specific to the type of wastes that are of interest here, the Plan Nacional de Residuos de Construcción y Demolición 2001–2006 (PNRCD) [National C&DW Plan] (BOE 12 July 2001), which approached the treatment, recovery, and recycling of these wastes. These national plans were substituted by the Plan Nacional Integrado de Residuos (PNIR) [National Integrated Waste Plan] for the period 2008–2015 (BOE 26 February 2009), within which, under Point nº 12, it covered the second National C&DW Plan. Among the plans’ objectives was the controlled collection and proper management of 95% of C&DW by 2011, the reduction or reuse of 15% of C&DW by 2011, the recycling of 40% of C&DW by 2011, and the exploitation of 70% of all waste construction material packaging, as from 2010.

Among other aspects, the RD 105/2008 introduced the obligation of including a C&DW management study in a building or construction work design project, which had to contain, as a minimum, an estimation of the amount of waste that could be produced, as well as measures for risk prevention, management process, and the exploitation of those waste streams.

Added to the current situation is an increasing demand for fines linked to the environmental restrictions on the exploitation of new quarries that have led to proposals for the reuse of certain waste streams as alternative raw materials. Different studies on the viability of valorizing granite waste and other ornamental rocks have been consulted. In this case, it is the waste from hornfels considered [6,7,8,9].

The renewal of a train track is carried out when the ballast does not meet the precise specifications related to wear and alteration of the rocks, which depends on their nature and is controlled by convoys that partially remove and replace the ballast. This control is performed when needed, according to the auscultator train.

In the present work, the addition of a construction waste stream from used ballast is studied for its use as a pozzolan. The need to activate this residue has been assessed using the pozzolanicity test, and the mechanical properties of the mixtures have been obtained by the partial replacement in the Ordinary Portland Cement (OPC) of the waste considered.

## 2. Materials

The aim was to obtain C&DW from discarded ballast that had been replaced by new ballast. To do so, the process of ballast wear was simulated using an accelerated method, which consisted of wearing down new ballast in a ball mill and collecting the fines. Aggregate materials were employed in this work, with coarse granulometry (gravel) from CANTERAS and CONSTRUCCIONES S.A. (CYCASA), at Aldeavieja, Ávila (Spain). The ballast was taken from the same batches supplied by the quarry for the renewal of the stretch of rail from PB Río Duero - Est. Valladolid C.G., on the Madrid—Segovia—Valladolid High Speed Rail Line.

The sample under study (C) is a hornfels (metamorphic rock) with granoblastic texture, although in many examples the regional schistosity can also be seen. Hornfels is dark rock with a matt shine and opaque colors.

An Ordinary Portland Cement (OPC) type CEM I 42.5 R cement was used by Italcementi group.

## 3. Methods

The Los Angeles abrasion test [10] was used to monitor wear, in accordance with the conditions specified in annex C of standard UNE ES 13450 [11].

The ballast fragments were worn down in a steel ball mill to obtain the waste product, that was then transported to a laboratory for oven drying for 24 h at 110 °C, to a constant weight (through the removal of humidity).

In accordance with the consulted literature [12,13], the calcination process was followed once the waste had been dried in an electric oven at 600 °C for 2 h (sample CC), with a view to establish the pozzolanic potential of this addition and to improve the activation potential of the waste.

To determine the pozzolanic activity, an accelerated method was used that consisted of placing 1 g of the sample solution in contact with 75 mL of a saturated solution of calcium hydroxide (17.68 mM/L) to 40 °C over 1, 7, 14, 28, and 90 days. The solution underwent vacuum filtration in a Buchner funnel at each age under study. The concentration of calcium ions expressed as calcium oxide or fixed lime was determined in the filtrate by the assay detailed in standard UNE—EN 196-5 [14,15].

The specific surface of the waste was studied using the BET method, through isothermal absorption of nitrogen, and the distribution of particle size (Sympatec Helos 12LA laser diffraction spectrometer, Sympatec, Clausthal-Zellerfeld, Germany).

The solid waste was analyzed through X-ray Fluorescence Spectroscopy (XRF) (to analyze chemical composition, using a Bruker S8 Tiger XDR spectrometer (Bruker, Fremon, CA, USA) with Spectra Plus Quant Express software v1.0.0.13), X-ray Diffraction (XRD) (to analyze mineralogical composition with a SIEMENS D-5000 diffractometer (Anton Parra, Madrid, Spain), working between 3 and 60 degrees with a sweep velocity of 2 degrees per minute), and scanning electron microscopy/energy dispersive X-ray spectroscopy (SEM/EDX) (providing surface data at a microscopic level and surface analysis, using a PHILIPS XL 30 flexible scanning electron microscope (Philips, Leuven Belgié) with a wolfram filament, a BIO-RAD SC 502 disk type sputter target, and an EDAX Energy Dispersive X-ray spectrometer with a DX4i silica/lithium detector and analyzer for chemical analysis, Philips, Leuven, Belgie).

Tests have been carried out on the samples for their resistance to compression and bending according to the UNE EN 196-1 standard [16] (compressive and flexural strength).

## 4. Results and Discussion

The particle size distribution of the discarded ballast waste was studied, showing a bimodal distribution with two maximum particle sizes from 6 to 15 µm.

BET surface determination provided a value of 1.32 m^2^/g, similar to the value obtained for fly ash of 1.40 m^2^/g, lower than the value for ceramic tiling of 3.00 m^2^/g [17], very much lower than fly ash at 20 m^2^/g [18], and higher than the values close to 0.98 m^2^/g for ladle furnace slag [18].

The result of the chemical analysis by XRF of the original sample (C) of OPC cement is provided in Table 1.

In addition, concentrations of zirconium, copper, chrome, cobalt, nickel, strontium, vanadium, zinc, and lead were detected in quantities that were not in excess of 50 ppm. The high content of both silica and aluminum points to good pozzolanic activity.

In turn, the mineralogical composition of the discarded ballast waste, obtained by XRD, indicated the presence of quartz, potassium feldspar, plagioclase (soda-lime feldspar), biotite, and clay-type minerals, such as kaolinite and chlorite with scarce little muscovite, as well as small quantities of hematite, as shown in Figure 1. The components of the cement were tricalcium aluminate, belite, alite, calcite, and ferrite phases.

### 4.1. Calcination of Ballast Waste

The waste material was calcined at a temperature of 600 °C for 2 h, in stove at a constant heat to increase its pozzolanic activity. The increase of this activity was due to the loss of structural water in the clay (kaolinite) minerals and in the phyllosilicates (biotite, chlorite).

The result of the mineralogical analysis of the calcined sample (CC) obviously affected the dehydroxylation of the kaolinite, which changed into amorphous metakaolinite, and had an incipient effect on the phyllosilicate structure, which started to lose hydroxyl groups. All other mineral components remained unchanged in the diffractogram.

### 4.2. Sample Pozzolanicity

The pozzolanicity test on the initial sample and the sample that had been thermally activated at 600 °C/2 h is shown in Figure 2. The improved behavior of sample C and the initial waste is shown, in all cases, except for at 28 days. Thermal activation of the waste was therefore not considered necessary and, henceforth, the initial sample was the only material used in the tests. The differences are so small that thermal activation is not required.

### 4.3. Preparation of Mortars with Discarded Ballast Waste

Standardized CEN sand was used with a granulometry of between 1 and 0.08 mm, which met the requirements specified in standard UNE EN 196-1 [16]. The cement was a CEM 1 42.5 R-type OPC, the composition of which is shown in Table 1 [19].

A mixture of the above components, ballast waste, and Portland cement was used to prepare a mixture that guaranteed the homogeneity of the corresponding mixtures. The cements were differentiated by the substitution of either 10% or 20% by weight of OPC for discarded ballast waste, in an attempt to design type II/A (6/20%) and IV/A (11–35%) cements, in accordance with standard UNE EN 197-1 [14].

### 4.4. Mechanical Behavior

Compressive strength and flexural tests were performed in accordance with standard UNE EN 196-1 [16]. Prismatic mortar specimens were prepared, measuring 4 cm × 4 cm × 16 cm, with a sand/cement ratio and water/cement ratio of 3/1 and 3/2, respectively. At 24 h after their manufacture, the specimens were demolded and cured, at a temperature of 20 ± 1 °C and a relative humidity of 100%, up until failure.

#### 4.4.1. Mechanical Strength under Compression

All the cements under study presented compressive strengths of over 10 MPa after two days of curing, and greater than or equal to 42.5 MPa over the following 28 days, thereby complying with the mechanical specifications contained in standard UNE EN 197-1 [14] for cements classed as high-strength (42.5 MPa).

From the graph shown in Figure 3, it may be seen that the incorporation of discarded ballast waste in the mortars in no way modified the existing logarithmic tendency (although the trend seems linear, the adaptation to a logarithmic equation meets R^2^ better) between compressive strength and curing time, regardless of the substitution level (10% or 20%). Correlation coefficients higher than 0.90 were obtained for R^2^.

The following expression was used to arrive that figure:y_OPC_ = 6.18 ln(x), + 41.64, con R^2^ = 0.945
while the two levels of substitution were calculated as follows:y_10%+90%OPC_ = 6.39 ln(x) + 34.88, with R^2^ = 0.984
y_20%+80%OPC_ = 6.24 ln(x) + 28.23, with R^2^ = 0.992

It was also noted that the incorporation of the discarded ballast waste implied a significant improvement in performance as the percentage substitution level increased. The weakened performance was close to 11% at 10% C + 90% OPC and around 23% at 20% C + 80% OPC, with regard to the standard OPC specimen, considering a time of 28 days of curing. This tendency to lose strength coincides with the trend noted in conventional mortars by Ramos et al. [20].

In addition, this behavior coincides with the behavior described by Frías et al. [21] and Vardhan et al. [22] in their studies on slate and marble quarry sludge, respectively, in the manufacture of new cements. An addition of 20% seems to improve the conditions of compressive strength compared to one of 10%.

The incorporation of different proportions of SiO_2_ and Al_2_O_3_ improved the structure of the pores, since the ballast waste affects the hydration of the cement and is responsible for the mechanical properties [23]. Fine pores are generated when the Si/Al ratio is high, which increases contact points and resistance. This mechanism is similar to secondary hydration and requires long ages to make it happen.

#### 4.4.2. Flexotraction Strength

Regarding the flexotraction strength of the mortars containing substitutions of ballast waste, a similar tendency was detected to the one observed for compressive strength, with a loss of strength of 10% and 17%, after 28 days of curing in the mixtures with substitution levels of 10% and 20%, respectively, as seen in Figure 4. In this case, the addition of 10% was more favorable in terms of flexotraction than the addition of 20% over time. It can be attributed to the delayed onset of the pozzolanic reaction. It is known that when the pozzolanic reaction begins, the amount of Ca (OH)_2_ decreases and the microstructure improves, with densification exceeding longer ages [24,25].

### 4.5. Total Porosity and Pore-Size Distribution

Table 2 shows the values of total porosity and the average pore size in the mortars with substitutions of ballast waste at 2 and at 90 days of curing. It is understood that the incorporation of ballast waste generates a slight increase in the total porosity of the mortars with substitutions of 10 and 20%, with a slightly higher increase at substitution levels of 10%, but with values very close to 9% with respect to OPC.

Table 2, likewise, shows the evolution of the average pore size, with a refinement of the system of pores as the cement hydration process progresses, showing a smaller average size with the age of curing at the higher substitution level (OPC, 10% and 20% of substitution), with respect to the mortars at 2 days.

The above is clear from the SEM images of the mortars under study. Accordingly, SEM-EDAX observations of the grain edges of the mortar specimens (4 cm × 4 cm × 16 cm) with substitution levels of ballast waste at 0%, 10% and 20%, cured over 90 days, have shown that when the cement has no substitution—as shown in Figure 5A—the inter-grain contact is formed by calcium silicate hydrate (CSH) gels and ettringite fibers that give it the slightly porous appearance that can been seen in Figure 5B. In turn, as demonstrated in Figure 5C,D, the inter-grain contact was not observed to be so well defined in the cement with no ballast waste. Nevertheless, as shown in Figure 5E–H, the tendency at levels of substitution of 10% were similar at 20%, although less acute, thus the inter-grain contacts were better defined and substitution levels at 20% were therefore considered unnecessary. The consideration of the results of the previous tests means that the addition is directed towards 10%, which would lead it to be considered a cement type II/A (6/20%) instead of the budget type IV/A (11–35%) in accordance with standard UNE EN 197-1 [14].

## 5. Conclusions

Discarded ballast waste has been used as a pozzolanic addition in cement, to reuse the C&DW instead of sending it to landfill. This type of waste participates in the Circular Economy due to its high amount of silicates, and, when using C&DW, it will be considered as a secondary raw material, as the displacement to waste plants or landfills is unnecessary when taking advantage of its reuse through its management in situ, with consequent economic benefits.

The use of this waste in the same type of facilities eliminates the participation of the necessary track material in its renewal.

The waste does not need any type of activation, neither thermal nor chemical, and can be directly used in mortar mixtures.

The importance of quantifying the addition to the OPC has been investigated at 10% and 20% for ballast waste. Both additions have given good results, especially 10%, for design type II/A (6/20%) cement. In the case of opting for the 20% addition, a type IV/A (11–35%) cement would be obtained, giving good results over time.

The variety of rocks used as ballast warrants further study, looking toward the consideration of materials other than hornfels.

## Figures and Tables

**Figure 1 materials-12-03887-f001:**
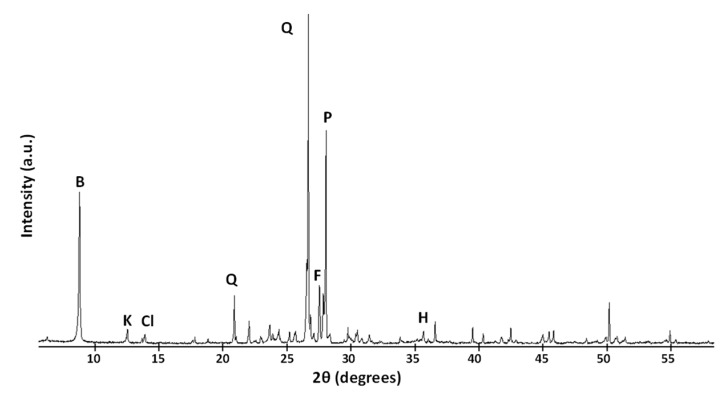
X-ray diffraction by the ballast waste (B = biotite; K = kaolinite; Cl = chlorite; Q = quartz; F = K feldspar; P = Ca, Na feldspar; H = hematite).

**Figure 2 materials-12-03887-f002:**
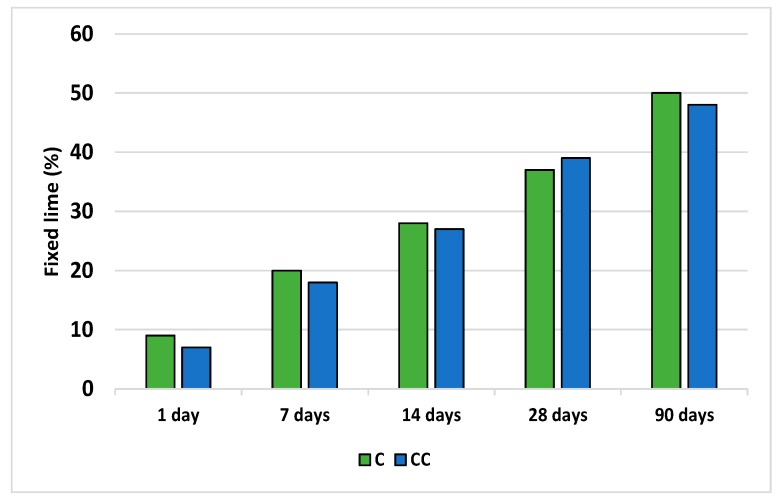
Measures of pozzolanicity of the initial discarded ballast waste (C) and the thermally calcined waste (CC).

**Figure 3 materials-12-03887-f003:**
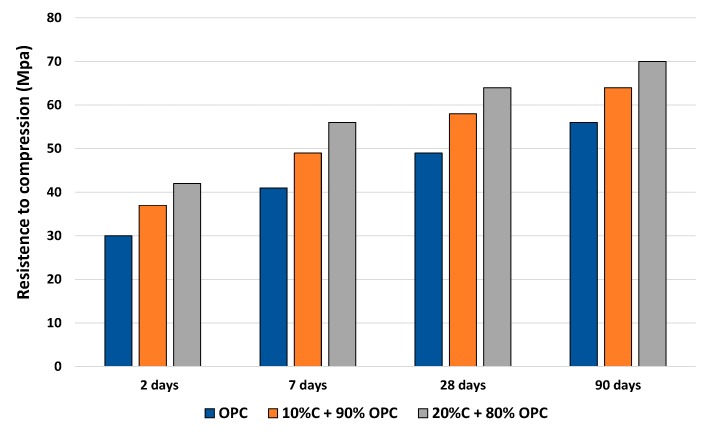
Variation of the compressive strength of the different mortars.

**Figure 4 materials-12-03887-f004:**
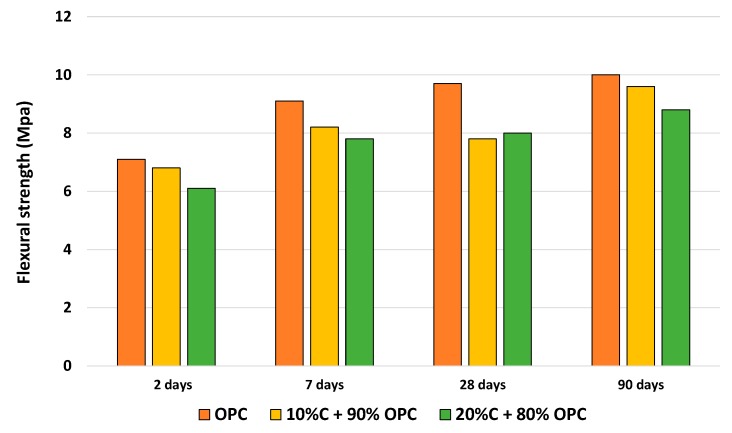
Variation of flexotraction strength of the different mortars.

**Figure 5 materials-12-03887-f005:**
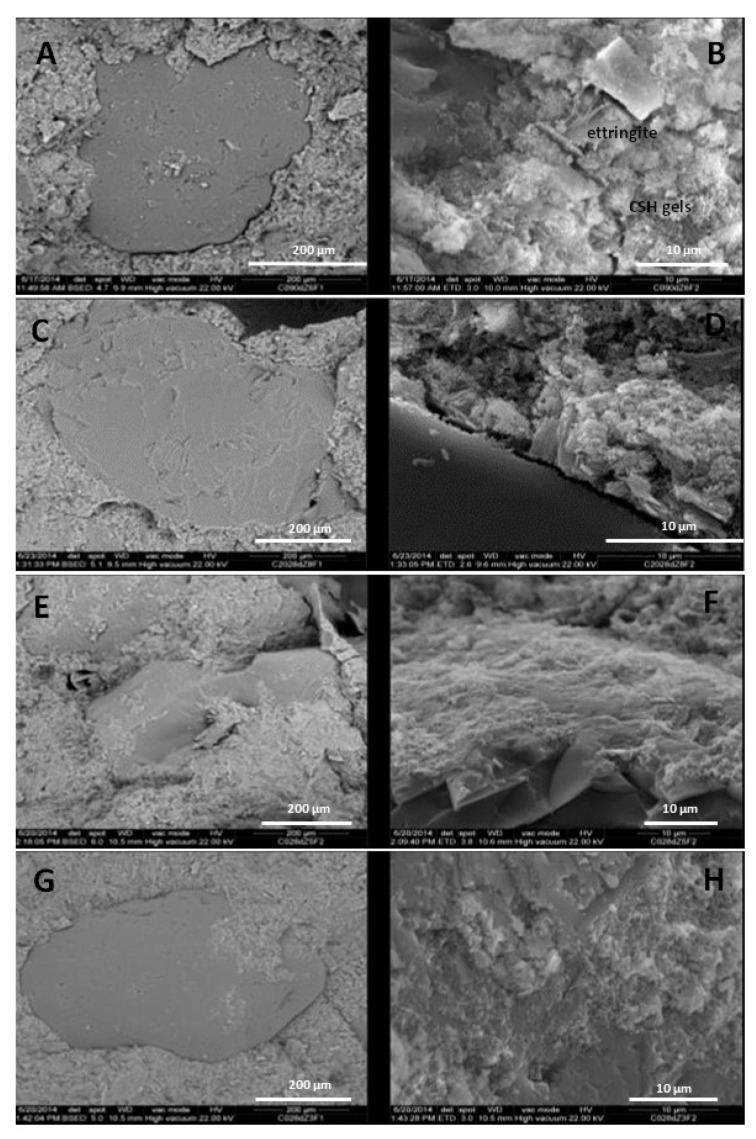
OPC Mortar specimen with no ballast waste: (**A**) magnified image of inter-grain contact; (**B**) grain edge; (**C**) magnified image of inter-grain contact of mortar specimen 20% + 80% OPC; (**D**) grain edge of mortar specimens 10% + 90% OPC; (**E**,**G**) magnified image of inter-grain contact; (**F**,**H**) grain edge.

**Table 1 materials-12-03887-t001:** Chemical analysis by X-ray Fluorescence Spectroscopy (XRF) from C sample and Ordinary Portland Cement (OPC) cement.

Oxides (%)	C Sample	OPC Cement
SIO_2_	69.64	20.26
Al_2_O_3_	15.00	4.61
Fe_2_O_3_	2.52	2.44
MgO	1.60	3.35
Na_2_O	3.59	4.14
K_2_O	4.04	1.41
P_2_O_5_	0.17	0.22
TiO_2_	0.51	0.14
MnO	0.04	0.03
Chloride (ppm)	290	130
LOI (loss on ignition)	0.52	3.04

**Table 2 materials-12-03887-t002:** Values of the total porosity and the pore diameter average at 2 and 90 days of curing for the mortars studied.

Mortar	Total Porosity (% vol.)		Pore Diameter, Average (µm)	
	2 Days	90 Days	2 Days	90 Days
OPC	13.02	11.98	0.0971	0.0732
10% + 90% OPC	14.52	12.16	0.0989	0.0721
20% + 80% OPC	14.30	12.81	0.1025	0.0742

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
