# Peer review of "Reusing Discarded Ballast Waste in Ecological Cements"

_materials, 2019, doi:10.3390/ma12233887_

Round 1
Reviewer 1 Report
The paper addresses an actual and very interesting topic. The use of discarded ballast waste to add to cement can certainly contribute to reducing the amount of waste to be sent to landfills and therefore to reduce environmental pollution.
In general, the paper is well structured. The methodology is appropriate. The bibliography is adequate.
The introduction provide sufficient background, but it can be improved.
In line 35 there is a mistake: "Puzzolini" must be corrected with "Pozzuoli", a city near Naples.
The "Methods" and "Results and discussion" sections are sufficiently detailed.
The "Conclusions" section is instead too concise and don’t emphasize the originality of the study. Therefore the authors are invited to better develop this section also highlighting:
- how the benefits related to the use of discarded ballast waste as additives of cements can be evaluated;
- what are the research perspectives.
Furthermore, an economic evaluation of costs and benefits should be added.
Author Response
Reviewer 1:
We welcome the suggestions of the reviewers that have been added to the final text and that have improved the work.
The corrections are highlighted in the text in red.
Thank you very much for your comments that have improved the better understanding of the work.
On line 35, Puzzolini by Pozzuoli has been changed.
The conclusions have been rewritten taking into account your comments.

Reviewer 2 Report
Specific items to correct and clarify:
20: Use the acronym CDW after first use of Construction and Demolition Waste.
23: Why you substitute 10%-20% and in line 17 you say that the rage is 9-15%?.
26-29: This lines are repeated in the abstract (20-23).
36: Include a convenient reference.
40: Include a convenient reference.
43: Include a convenient reference.
50: Include a convenient reference.
61: The PNRU and the PNRCD are dated in 2001 and 2002. How can they culminate the RD 105/2008 which was published seven years later?.
86: Finish the introduction point with a single paragraph explaining what is the general structure of the paper and what is going to be said in every single point (briefly).
88: Explain (in the Introduction point), in why and in which cases an old ballast is replaced with a new one, and how often it happens.
98: The acronym OPC has to be explained before it first use in the text (as it is in line 133). Please include a final table after Bibliography with the definition of all the acronyms used in the text.
180: Explain why the tendency is logarithmic, because it seems to be quite linear
190: Figure 2 shows an improved resistance to compression for mortars with C additions than other just 100% OPC. That does not concorde with the asseveration in the text
237: Figure 3, and Figure 2 (according to what is said in the text) shows that the performance of the compression and flexotraction strength of the mortar is lower that the one without OPC. Then explain why you make that conclusion.
237: Need to improve and enrich the conclusions of the paper to make it relevant enough to be considered for publication
246: Consider to include some relevant references from Materials MDPI Journal
Author Response
Reviewer 2:
We welcome the suggestions of the reviewers that have been added to the final text and that have improved the work.
The corrections are highlighted in the text in red.
Specific items to correct and clarify:
20: Use the acronym CDW after first use of Construction and Demolition Waste
It has been changed in lone 20 and others lines
23: Why you substitute 10%-20% and in line 17 you say that the range is 9-15%
Sorry, in this paper, the replacement is 10 and 20%. It has been corrected.
26-29: This lines are repeated in the abstract (20-23).
This lines are deleted.
36, 40, 43 and 50. Include a convenient reference.
Have been added.
The PNRU and the PNRCD are dated in 2001 and 2002. How can they culminate the RD 105/2008 which was published seven years later?
Indeed PNRU and PNRCD are RD precursors. It has been corrected in the text.
Finish the introduction point with a single paragraph explaining what is the general structure of the paper and why is going to be said in every single point (briefly).
The end of the introduction has been rewritten.
Explain (in the Introduction point), in why and in which cases an old ballast is replaced with a new one, and how often it happens.
It has been explained.
The acronym OPC has to be explained before it first use in the text (as it is in line 133). Please include a final table after Bibliography with the definition of all the acronyms used in the text.
It has been corrected and the table has been included.
Explain why the tendency is logarithmic, because it seems to be quite linear.
Although the trend seems linear, the adaptation to a logarithmic equation meets R2 better.
Figure 2 shows an improved resistance to compression for mortars with C additions than other just 100% OPC. That does not concorde with the asseveration in the text.
Has been corrected.
Figure 3, and Figure 2 (according to what is said in the text) shows that the performance of the compression and flexotraction strength of the mortar is lower that the one without OPC. Then explain why you make that conclusion.
In mixtures the flexotraction is greater with time it differs little. In Figure 3 OPC and 10% substitution are almost the same values.
Need to improve and enrich the conclusions of the paper to make it relevant enough to be considered for publication.
The conclusions have been changed and rewritten.
Consider to include some relevant references from Materials MDPI Journal.
Two references on the subject have been included.
Khan, K.; Amin, M.N.; Saleem, M.U.; Qureshi, H.J.; Al-Faiad, M.S.; Qadir, M.G. Effect of Fineness of Basaltic Volcanic Ash on Pozzolanic Reactivity, ASR Expansion and Drying Shrinkage of Blended Cement Mortars. Materials 2019, 12 (16), 2603-2612.
Velazquez, S.; Monzó, J.M.; Borrachero, M.V.; Payá, J. Assessment of the Pozzolanic Activity of a Spent Catalyst by Conductivity Measurement of Aqueous Suspensions with Calcium Hydroxide. Materials 2014, 7 (4), 2561-2576.

Reviewer 3 Report
Abstract is too long and reads like an introductory paragraph, not a succinct summary of the paper.
Introduction: provides interesting, detailed information on the context.
Lines 79-80 etc I did not clearly understand the source of the material or what industrial process was generating the waste. “Used ballast” from what process? What kind of ballast?
Lines 95-96 The two mineral names are not familiar to me (perhaps my ignorance): a check suggests cornubite is a rare copper arsenate mineral. Is that right? Perhaps more detail and references here? Corneine I could not find.
The material analysis (SEM, XRF, XRD etc) is fairly comprehensive and appropriate. The technical testing (pozzolanicity etc) is likewise appropriate.
The references could perhaps take more account of the wide literature on the use of industrial mineral wastes as secondary cementing materials.
Minor query: line 91 “thick” granulometry, not clear meaning (= coarse?)
Comment
The paper is written in excellent English. In my view it is publishable, although not likely to be of wide interest. I would suggest rewriting a proper abstract, and moving much of the existing abstract into the Introduction. The Abstract should be a brief summary of the paper. The main text would be of greater interest if the work could be placed and assessed in relation to what is known more widely about waste-stream pozzolans; in the absence of this, the paper is really just a technical assessment of one single material at one single location.
Author Response
Reviewer 3:
We welcome the suggestions of the reviewers that have been added to the final text and that have improved the work.
The corrections are highlighted in the text in red.
Abstract is too long are reads like an introductory paragraph, not a succinct summary of the paper.
The abstract has been shortened and changed.
Lines 79-80 did not clearly understand the soured of the material or what industrial process was generating the waste:
The residue is from the wear of the road ballast caused by the passage of the trains that round the fragments, in addition to their weathering.
Lines 95-96. The two mineral names are not familiar to me…….
The names of the rock have been changed and a description of it has been incorporated.
The references could perhaps take more account of the wide literature on the use of industrial mineral waste as secondary cementing materials.
Four references that enrich the study have been added in the introduction,
Line 91 thick granulometry not clear meaning.
The word has been replaced.

Reviewer 4 Report
The paper investigated the substitution of cement by discarded ballast, which is attractive. However, the presentation of paper is more like a report. Need more scientific design and findings for paper publication.
1. Introduction is weak without showing logic of experimental design and lack of relevant studies.
2. Did not show mechanical tests in Methods.
3. No XRD results.
4. I do not see any discussion that support the last conclusion "the substitution level at 10% to prepare type IV (pozzolanic) cements". I expect to see why using discarded ballast waste is comparable to traditional type IV cements.
5. Mix proportion is not provided. Instead of OPC, better to use a traditional type IV cement as control set.
6. Discussion only presents results without any mechanism or depth.
Author Response
Reviewer 4:
We welcome the suggestions of the reviewers that have been added to the final text and that have improved the work.
The corrections are highlighted in the text in red.
Introduction is weak without showing logic of experimental design and lack of relevant studies.
Consulted the bibliography no references to the application of ballast residues have been found. References have been incorporated on the use of granite and marble ornamental rock waste.
Did not show mechanical tests in Methods.
The methods of mechanical tests according to UNE EN 196-1 standard have been added.
No XRD results.
A diffractogram of the ballast waste has been added (Figure 1).
I do not see any discussion that support last conclusion “the substitution level at 10% to prepare type IV (pozzolanic) cements. ". I expect to see why using discarded ballast waste is comparable to traditional type IV cements.
The reason for comparing with a type II / A cement has been incorporated into the results.
Mix proportion is not provided. Instead of OPC better to use a traditional type IV cement as control set.
The proportion of the addition is 10 and 20%, the first one being more favorable.
Discussion only presents results without any mechanism or depth.
Results discussion is completed.

Round 2
Reviewer 2 Report
Fix the mistake in line 329
Author Response
Reviewer 2.
Once again thank you very much for your wise suggestions. The corrections will go green
Fix the mistake in line 329.
Sorry. ED by EU has been changed, also on line 49.

Reviewer 3 Report
The paper is now considerably improved, and in my view can be published.
Author Response
Reviewer 3.
Once again thank you very much for your wise suggestions.
Thanks.

Reviewer 4 Report
The entire discussion only addressed the trends or results you obtained, without any explanations on the reasons behind them.
Author Response
Reviewer 4:
Once again thank you very much for your wise suggestions. The corrections will go green
The entire discussion only addressed the trends or results you obtained, without any explanations on the reasons behind them.
Paragraphs have been added in discussion with new references.
